# Dual Strategy Based on Quantum Dot Doping and Phenylethylamine Iodide Surface Modification for High-Performance and Stable Perovskite Solar Cells

**Shulan Zhang [1]**, **Renjie Chen [1]**, **Mujing Qu [1]**, **Biyu Long [1]**, **Nannan He [1]**, **Sumei Huang [1,2]**, **Xiaohong Chen [1,2,\*]**, **Huili Li [1,2,3,\*]** and **Tongtong Xuan [4,\*]**

1. Engineering Research Center for Nanophotonics & Advanced Instrument, Ministry of Education, School of Physics and Electronic Science, East China Normal University, Shanghai 200241, China; mjing2021@163.com (M.Q.); shihnna@163.com (N.H.)
2. Joint Institute of Advanced Science and Technology, East China Normal University, Shanghai 200241, China
3. Chongqing Key Laboratory of Precision Optics, Chongqing Institute of East China Normal University, Chongqing 401120, China
4. Fujian Key Laboratory of Surface and Interface Engineering for High Performance Materials, College of Materials, Xiamen University, Xiamen 361005, China
* Correspondence: xhchen@phy.ecnu.edu.cn (X.C.); hlli@phy.ecnu.edu.cn (H.L.); ttxuan@xmu.edu.cn (T.X.)

**Abstract:** High-quality perovskite films (PFs) are crucial for achieving high-performance perovskite solar cells (PSCs). Herein, we report a dual-modification strategy via incorporating $CsPbBr_3$ QDs into $MAPbI_3$ perovskite bulk and capping the interface of the perovskite/hole transport layer (HTL) with phenylethylamine iodide (PEAI) to improve perovskite crystallinity and interface contact properties to acquire high-quality PFs with fewer defects. $CsPbBr_3$ QDs promoted perovskite grain growth and reduced bulk defects, while PEAI surface modification passivated interfaces, improved hydrophobic properties, and prevented carrier recombination at the perovskite/HTL interface. Benefiting from growth control and the effective suppression of both bulk and interface carrier recombination, the resulting devices show a greatly improved photoelectric conversion efficiency (PCE) from 17.21% of the reference cells to 21.04% with a champion Voc of 1.15 V, Jsc of 23.30 mA/cm$^2$, and fill factor (FF) of 78.6%. Furthermore, the dual-modification strategy endows PFs with promoted moisture tolerance, and the nonencapsulated PSCs retain 75% of their initial efficiency after aging for 30 days at 40% relative humidity and room temperature, which is significantly higher than the 59% value of the original PSCs. Good operational stability and the maintained efficiency of the target device over time suggest the potential for future commercialization.

**Keywords:** perovskite solar cells; doping $CsPbBr_3$ QDs; PEAI surface modification; stability

## 1. Introduction

Organic–inorganic hybrid perovskite solar cells (PSCs) have become the most promising next-generation photovoltaic technology due to their advantages of a simple preparation process, low cost, adjustable direct bandgap, and high photoelectric conversion efficiency (PCE) [1–3]. In 2009, the first PSCs were successfully prepared by using methylammonium lead iodide ($MAPbI_3$) and methylammonium lead bromide ($MAPbBr_3$) perovskite material as a light-absorbing layer and achieved a PCE of 3.8% [4]. Three years later, Michael Gratzel's group introduced the solid-state Spiro-OMeTAD (2,2′,7,7′-Tetrakis [N,N-di (4-methoxyphenyl) amino]-9,9′-spirobifluorene) as the hole transport layer (HTL) material, which made the PCE of PSCs improve to 9.7% [5]. Subsequently, more and more researchers have engaged in this field, and the performance of PSCs has constantly been refreshed by various technologies [6,7]. At present, the latest certification efficiency of PSCs has reached 26.1% [8], which is very close to the conversion efficiency of monocrystalline silicon solar cells.

As the crucial part of PSCs, the crystallinity, grain size, defect density, and interfacial characteristics of the perovskite light-absorbing layer will significantly affect the light absorption, stability, carrier transport and extraction, PCE, and service life of the final PSCs [9–11]. Therefore, high-quality perovskite films (PFs) with well-crystallized grains, large grain size, high surface coverage, and low surface roughness are considered to be essential to acquire high-performance PSCs [6,12,13]. Previous investigations have demonstrated that PFs with the desired morphologies can be fabricated by the anti-solvent process [14–16], dual-source co-evaporation [17], spraying method [18], and so on. Among them, dripping anti-solvent into the perovskite precursor is an effective "one-step" solution process for preparing highly efficient PSCs, particularly when adopting cesium lead halide perovskite quantum dot ($CsPbBr_3$ QD) additives into the anti-solvent to control nucleation and further improve the quality of PFs [19,20]. As is known, $CsPbBr_3$ QDs have demonstrated swift achievements for potential display applications because of their excellent optical properties, such as ultrahigh photoluminescence quantum yield (PLQY), tunable bandgap, narrow full width at half maximum (FWHM), high defect tolerance, etc. [21]. Based on these, Yao et al. added $CsPbBr_3$ QDs to Diethyl Ether (DE) anti-solvent as heterogeneous nucleation centers to promote the formation of smooth and pinhole-free PFs with increased particle size, reduced defects, and increased carrier lifetime. Finally, the champion PSCs showed a PCE of 20.17% [22].

Solution-processed PFs are polycrystalline, and usually contain many substantial defects such as grain boundaries, vacancies, lattice discontinuities, and surface microscopic pinholes. External water vapor and oxygen will first destroy perovskite crystals through these defects to cause irreversible phase transition and degradation of perovskite from the outside to the inside [23–25]. These defects are most easily formed on the surface and interfaces of PFs; therefore, the passivation of the surface and interface defects is always one of the most important tasks when fabricating highly efficient PSCs [26–28]. Recently, several halide organic amine salts such as butyl amine iodide (BAI), naprodium methamine iodide (NMAI), and phenylethylamine iodide (PEAI) have been used to modify two-dimensional PFs to passivate dangling bonds [29,30].

In summary, both surface modification and nucleation growth control through adding QDs to anti-solvents exhibit great promotion of highly efficient and stable PSCs. The dual-protection strategy achieved by combining them is a better choice to acquire high-quality PFs and ultimately improve the performances of PSCs. In this paper, a very small amount of $CsPbBr_3$ QDs were first added to ChloroBenzene (CB) anti-solvent to help the nucleation and grain growth of $MAPbI_3$ films. $Br^-$ and $Cs^+$ in $CsPbBr_3$ entered into the inner crystal lattice of $MAPbI_3$ to replace $MA^+$ and $I^-$ and form a dense and stable $Cs_xMA_{1-x}PbI_{3-y}Br_y$ film with low defects and large grains. Then, as-fabricated $Cs_xMA_{1-x}PbI_{3-y}Br_y$ was secondarily passivated by spin-coating an ultrathin PEAI molecular layer on its surface to reduce the surface roughness and further fill the surface and grain boundary defects. As a result, the non-radiative recombination between carriers at the perovskite layer surface was reduced and the leakage current inside the device was inhibited by hindering the direct contact between electrons and HTL at the grain boundary. Benefiting from the dual-protection strategy, the fabricated device with dual modification exhibited considerable efficiency enhancement from 17.21% of the reference to 21.04%. The hydrophobic property of the PEAI molecular layer combined with the dense and high-crystallinity $Cs_xMA_{1-x}PbI_{3-y}Br_y$ structure markedly enhanced the moisture resistance and stability of the jointly modified PSCs.

## 2. Experimental Materials and Methods

### 2.1. Materials

$Cs_2CO_3$ (99.5%), oleic acid (OA, 80%–90%), oleamide (OAm, 80%–90%), 1-octadecene (ODE, 90%), isopropyl alcohol (99.9%), N, N-dimethylformamide (DMF, 99.9%, HPLC), dimethyl sulfoxide (DMSO, 99.9%, ultra-dry), CB (99%) and bis(trifluoromethane)sulfonimide lithium salt (Li-TFSI, 99.9%) were purchased from Aladdin (Shanghai, China). Methylammonium iodide

(MAI, 99.5%), lead (II) iodide (PbI$_2$, 99.99%), and lead bromide (PbBr$_2$, 99%) were purchased from Xi'an poly company (Xi'an, China). Toluene (99.5%) was purchased from Chinese medicine (Shanghai, China). Ethylacetate (99.8%, water $\leq$ 50 ppm) was bought from Acmec (Shanghai, China). SnO$_2$ aqueous solution (Alfa, 15%) and Spiro-MeoTAD (99.5%) were bought from Ningbo Borun New Materials Company (Ningbo, Zhejiang, China).

### 2.2. Synthesis of CsPbBr$_3$ Perovskite QDs

Preparation of Cs-oleate (Cs-OA): 0.2 g Cs$_2$CO$_3$, 10 mL ODE, and 1.25 mL OA were added into a 25 mL 3-neck flask which was full of N$_2$. The mixture was continuously stirred with a magnetic stirrer. At the same time, the reaction temperature slowly rose to 120 °C for 1 h, and then the solution was heated to 150 °C for 5 h under N$_2$ until it was completely clear. The resultant Cs-OA was stored in N$_2$ and pre-heated to 100 °C before injection.

Synthesis of CsPbBr$_3$ QDs: 0.2 g PbBr$_2$ and 10 mL ODE, 1.5 mL OA, and 1.5 mL OAm were added to a 25 mL three-necked flask and slowly heated to 120 °C for 1 h under vacuum. After complete solubilization of a PbBr$_2$ salt, the solution was heated to 160 °C, and then 0.8 mL Cs-oleate precursor was rapidly injected. After complete reaction for 10 s, it was quickly cooled by ice bath treatment to obtain the CsPbBr$_3$ QDs. It was purified more than twice with a mixture of EA and toluene (2:1). Finally, as-prepared CsPbBr$_3$ QDs were dispersed in CB according to different concentration requirements for further use.

### 2.3. Device Fabrication

The etched ITO glass (10 $\Omega/\square$) was cleaned by detergent, deionized water, ethanol, and acetone, and blow-dried with clean air. The diluted SnO$_2$ colloid solution (3 wt%) was spin-coated on the ITO glass at the speed of 3000 rpm/min for 30 s and then annealed at 150 °C for 30 min. After annealing, the SnO$_2$ compact layer was cleaned using UV ozone before further spin-coating of the next layer. Next, 55 µL perovskite precursor (40 wt%, 1M MAI, 1M PbI$_2$ dissolved in DMF/DMSO (8:2)) was spin-coated on the SnO$_2$ layer at the speed of 3000 rpm/min for 30 s; 10 s after the start of the process, 300 µL CB with/without CsPbBr$_3$ QDs was dropped onto the wet film, and then annealed at 100 °C for 10 min.

For the PEAI-modified layer, 2 mg/mL of PEAI isopropanol solution was directly used for spin coating (4000 rpm/min) on the CsPbBr$_3$ QD-doped perovskite film layer for surface modification.

The Spiro-OMeTAD solution (72.3 mg/mL in CB) with 28.5 µL TPB and 17.5 µL of Li-TFSI solution (520 mg/mL in acetonitrile) was spin-coated on the perovskite layer at the speed of 4000 rpm/min for 30 s to form HTL. Finally, a 90 nm thick Ag electrode was evaporated under the vacuum pressure of $8 \times 10^{-4}$ Pa.

### 2.4. Characterizations

The transmission electron microscope (TEM) images were obtained by a high-resolution transmission electron microscope from Hitachi F 2100 (Tokyo, Japan). The UV-Vis absorption spectra were characterized by using a UV-vis spectrophotometer (Cary 5000), (Beijing, China). The steady-state and time-resolution photoluminescence (TRPL) spectra were recorded by the FLS 980 system (Edinburgh, UK). The X-ray power diffractometer (XRD) was tested by a Bruker D8 ADVANCE (Karlsruhe, Germany) using Cu K$\alpha$ radiation as the X-ray source. Scanning electron microscope (SEM) images were obtained by a Zeiss Sigma 300 system (Oberkochen, Germany). Atomic force microscopy (AFM) was performed using a Nanoscope V Multimode 8 scanning probe microscope from Bruker Corporation (Karlsruhe, Germany). Atomic energy-dispersive X-ray spectroscopy (EDS) mapping of the perovskite was taken by a JEM-2100 high-resolution transmission electron microscope (JEOL, Doakido, Japan). The decay curve was recorded by using the FLS 980 with the laser emission wavelength of 470 nm. X-ray photoelectron spectroscopy (XPS) data were tested by an ESCALAB 250XI system (Thermo Scientific Company, Waltham, MA, USA). The J-V curves and steady power output were tested by a Newport Sunlight Simulator(Newport Company, Dresher, PA, USA), Keithley-2440 digital resource and 0.1 cm$^2$ Shelter mask. The

electrochemical impedance spectra (EIS) curves were characterized by using an Auto-lab chemical station with the bias voltage of −0.9 V. The C-V, space charge-limited current (SCLC), transient photovoltage (TPV), and transient photocurrent (TPC) decay curves were obtained by using an electrochemical workstation (Zahner, Kronach, Germany). The stabilities of PSCs were characterized by recording their steady-state current outputs under 100 mW/cm$^2$ simulating sunlight, and normalized PCE curves varied with time at a relative humidity of 40% and room temperature (RT). It needs to be pointed out that experiments were carried out in an ambient atmosphere without encapsulation.

## 3. Results and Discussion

### 3.1. Morphology and Optical Properties of CsPbBr$_3$ QDs

To achieve the purpose of doping with CsPbBr$_3$ QDs to assist the nucleation and grain growth of PFs, we must first synthesize uniform and stable CsPbBr$_3$ QDs. Figure 1 shows the TEM morphology and PL spectra of as-prepared CsPbBr$_3$ QDs. It can be seen that as-synthesized QDs present uniformly dispersed cubic crystalline grains with an average diameter of about 10 nm (Figure 1a). Under the excitation of 365 nm ultraviolet (UV) light, it emits bright green fluorescence with an emission peak at 511.1 nm, which is consistent with the emission of CsPbBr$_3$ QDs reported in the references [31].

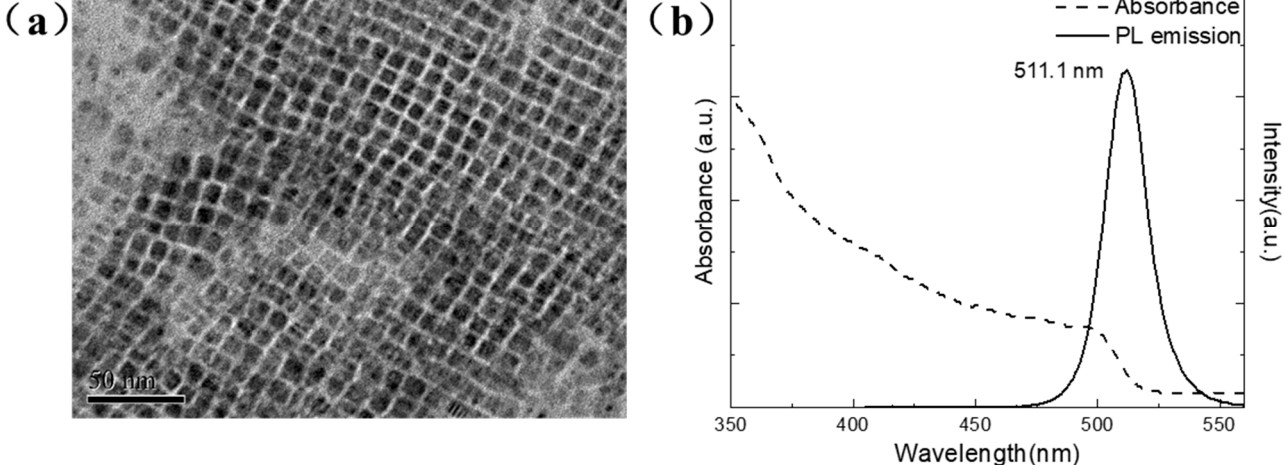

**Figure 1.** (**a**) TEM image and (**b**) PL spectra of as-prepared CsPbBr$_3$ QDs ($\lambda_{ex}$ = 365 nm).

In order to confirm that the cubic grain belongs to the CsPbBr$_3$ phase, XRD characterization was performed and is presented in Figure 2. The diffraction peaks at 15.9° and 31.3° are attributed to the (100) and (200) crystal planes of cubic CsPbBr$_3$ perovskite (PDF 18-0364), respectively, and no other impurity peaks appear. This proves that pure CsPbBr$_3$ QDs with good crystallinity have been successfully synthesized.

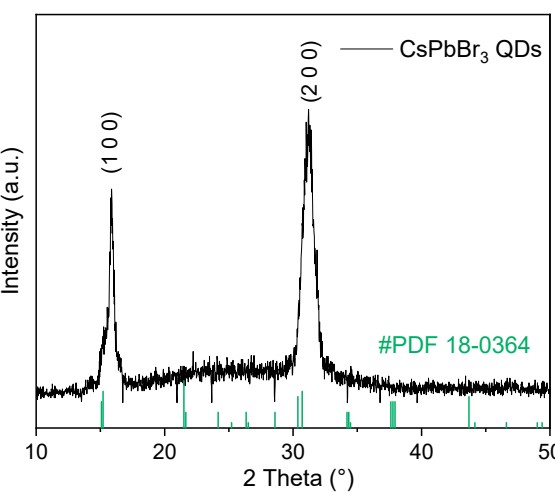

**Figure 2.** XRD pattern of as-prepared CsPbBr$_3$ QDs (PDF 18-0364).

### 3.2. Effect of CsPbBr$_3$ QDs Doping and PEAI Surface Modification on the Morphology and Crystallization of PFs

Figure 3a–d show the SEM images of MAPbI$_3$ films doped with different concentrations of CsPbBr$_3$ QDs at 0, 0.2, 2, and 10 μg/mL in CB anti-solvent. It can be seen that the grain size of the reference is relatively small, while after adding CsPbBr$_3$ QDs to the anti-solvent, MAPbI$_3$ crystals obviously grow and reach the maximum at 2 μg/mL CsPbBr$_3$ QDs. This is attributed to the CsPbBr$_3$ QDs present in the anti-solvent, which act as nucleation sites during the perovskite growth process. The appropriate concentration of CsPbBr$_3$ QDs can facilitate the nucleation process, leading to the formation of large-grain-size and dense perovskite films. However, the introduction of an excessive amount of CsPbBr$_3$ QDs nucleation sites can restrict the growth of MAPbI$_3$ perovskite crystals, leading to a reduction in grain size [19,32]. When concentration is further increased to 10 μg/mL, the grain size of the film is reduced instead, but it is still larger than that of the reference sample [32]. Due to the limitations of the resolution and SEM technique, there is no significant change in the SEM image of jointly modified PFs (Figure 3e). Therefore, more advanced AFM was adopted to characterize their three-dimensional surface morphologies. The results are exhibited in Figure S1.

The grain size of PFs doped with 2 μg/mL CsPbBr$_3$ QDs was significantly increased from 161.2 ± 25.4 nm in the control to 235.1 ± 22.1 nm, and simultaneously the surface became denser and smoother with a reduced surface roughness (RMS) of 65.1 nm (75.6 nm for the reference perovskite). After further surface modification by PEAI (Figures 3e and S1c), although the grain size of the film did not increase significantly (236.4 ± 23.1 nm), the RMS was further reduced to 58.8 nm. This reduction in roughness indicates that the modified PEAI molecules can uniformly cover the grain boundaries and interfaces of the perovskite layer, thereby finely tuning the surface morphology of the film [30].

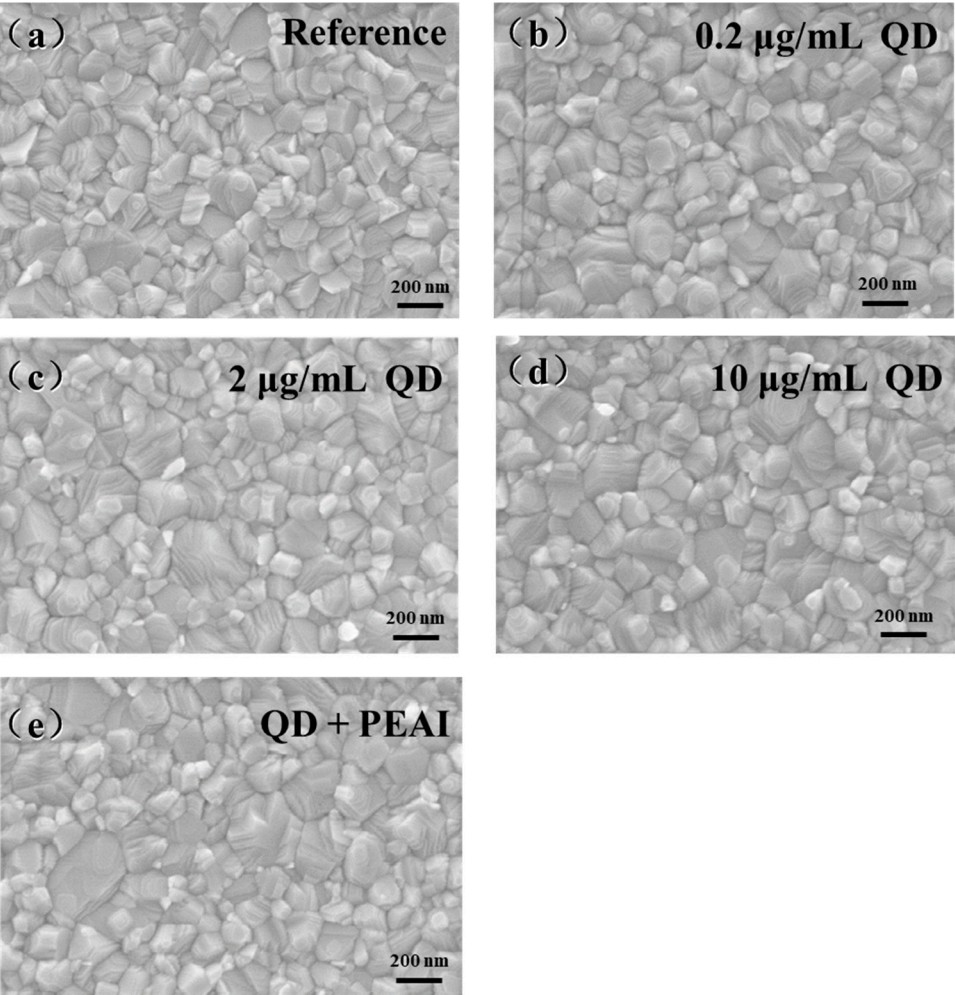

**Figure 3.** SEM morphologies of PFs fabricated under different modification conditions: (**a**) reference sample; (**b**) 0.2 μg/mL, (**c**) 2 μg/mL, and (**d**) 10 μg/mL CsPbBr$_3$ QD doping; (**e**) jointly modified with 2 μg/mL CsPbBr$_3$ QDs and 2 mg/mL PEAI.

The largest grain size for the CsPbBr$_3$ QD-modified perovskite layer is further supported by XRD patterns from Figure 4. The (110) peak intensity of PFs modified with CsPbBr$_3$ QDs was enhanced compared to the undoped sample. The PFs with 2 μg/mL CsPbBr$_3$ QDs showed the strongest peak intensity and the corresponding narrowest FWHM of 0.178°, as shown in Table S1. From Figure 4b and Table S1, it can also be found that the position of the (110) peak gradually moves towards the large angle from 13.97° for the reference to 14.04° of the PFs modified with 10 μg/mL CsPbBr$_3$ QDs. This is because the smaller radius of Cs$^+$ and Br$^-$ ions in CsPbBr$_3$ QDs enter the MAPbI$_3$ lattice to replace the larger MA$^+$ and I$^-$ to form Cs$_x$MA$_{1-x}$PbI$_{3-y}$Br$_y$, resulting in a contraction of the perovskite lattice [33], which can be further proved by EDS mappings of CsPbBr$_3$ QDs and PEAI dual-modification PFs in Figure S2. Br elements, like Pb and I, are evenly distributed in the grains of perovskite after further surface modification by PEAI (Figure S1c), although the grain size, (110) diffraction peak intensity, position, and FWHM remain almost unchanged (Table S1). This demonstrates that the modified PEAI molecules can uniformly cover the grain boundary and interface of the perovskite layer, slightly adjust the surface morphology of the film, and thereby reduce its roughness [34,35]. In addition, PEAI also has the function of filling the ion vacancies and passivating defects on the surface of PFs, which will be discussed in detail later. High-quality PFs with better crystallinity, large grain size, and a dense and smooth surface will help to improve the photovoltaic performance of PSCs [36,37].

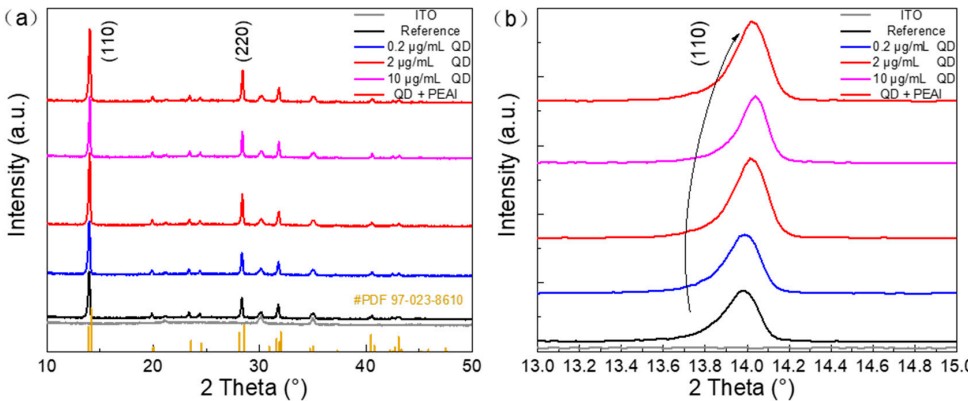

**Figure 4.** (**a**) The XRD patterns and (**b**) the enlarged (110) diffraction peaks of PFs modified with different materials (PDF 97-023-8610).

### 3.3. Effect of Joint Modification of CsPbBr₃ QDs Combined with PEAI on Optical Properties as Well as Binding Energy of PFs

The better crystallinity of the dually modified PFs can enhance their corresponding optical properties, as shown in Figure 5. As expected, the absorbance and PL intensity were gradually increased in the order of the reference sample, the PFs modified with only CsPbBr₃ QDs, and dually modified PFs based on QDs and PEAI (Figure 5a,c). However, the bandgap from Tauc plots and the emission position of the three kinds of PFs remain almost unchanged. These results support the superior crystallinity and reduced defects of the dual-modification MAPbI₃ films [36]. The better crystallinity of the perovskite layer usually produces a stronger PL intensity because fewer defects can reduce carrier recombination. The enhanced light absorption indicates the suppression of band-edge trap states by the dual-protection strategy, which will help to increase the photocurrent density of PSCs.

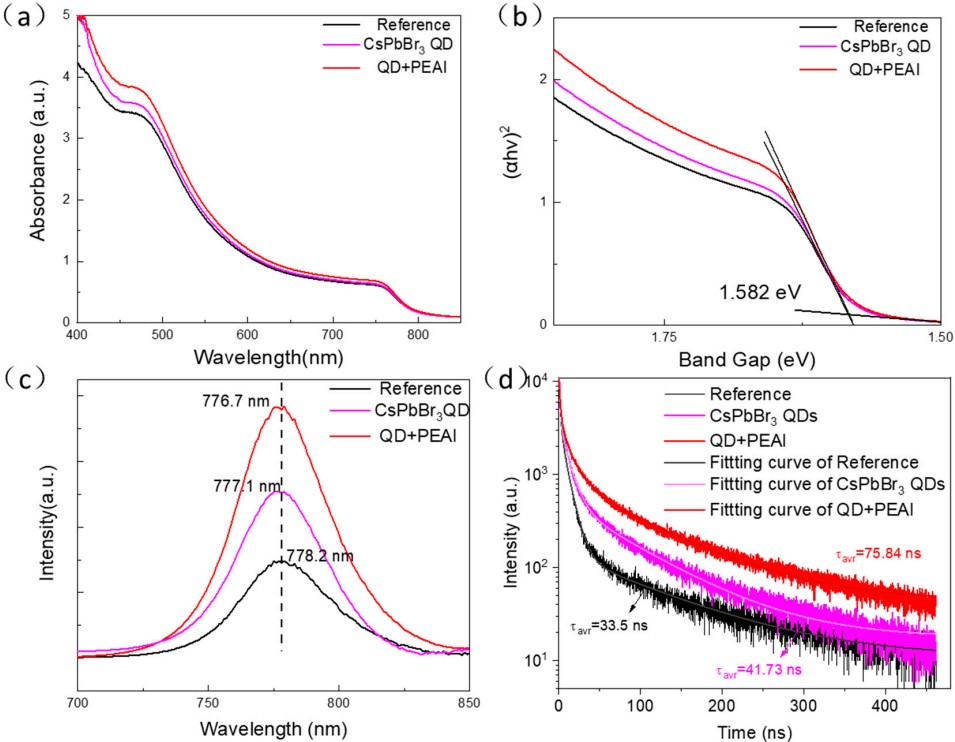

**Figure 5.** (**a**) Absorption spectra, (**b**) Tauc plots, (**c**) steady-state emission spectra, and (**d**) TRPL spectra of the reference sample, CsPbBr₃ QD-doped sample, and CsPbBr₃ QDs and PEAI jointly modified PFs on SiO₂ substrate.

To further study the charge-carrier recombination dynamic, TRPL spectra are exhibited in Figure 5d. The fluorescence decay curves can be fitted as a triple-exponential function like Equation (1) [38–40]:

$$Y(t) = A_1 e^{\frac{-\tau}{\tau_1}} + A_2 e^{\frac{-\tau}{\tau_2}} + A_3 e^{\frac{-\tau}{\tau_3}} \tag{1}$$

where $A_1$, $A_2$, and $A_3$ are the coefficients of the non-radiative, radiative, and Auger recombination, respectively, and $\tau_1$, $\tau_2$, and $\tau_3$ represent decay lifetimes [40]. The average fluorescence lifetime is calculated according to the following Equation (2):

$$\tau_{avr} = \frac{A_1 \tau_1^2 + A_2 \tau_2^2 + A_3 \tau_3^2}{A_1 \tau_1 + A_2 \tau_2 + A_3 \tau_3} \tag{2}$$

and they are 33.5 ns, 41.73 ns, and 75.84 ns, respectively, which is consistent with the strongest intensity of steady-state PL for the dually modified PFs. Undoubtedly, the longer lifetime is attributed to the high-quality film fabricated by the dual-protection strategy which passivates the defects in and on the surface of the perovskite layer and reduces the probability of non-radiative recombination between carriers. This will ultimately help to improve the $V_{oc}$ and FF of PSCs [37].

The comparative XPS spectra in Figure 6 can further verify the interactions between CsPbBr$_3$ QDs, PEAI, and MAPbI$_3$ perovskite crystals. Obviously, the binding energies of Pb 4f$_{5/2}$ and Pb 4f$_{3/2}$ of PFs dually modified with CsPbBr$_3$ QDs and PEAI increase from 138.45 eV and 143.3 eV of the reference perovskite layer to 138.63 eV and 143.47 eV, respectively, indicating that Pb$^{2+}$ in perovskite crystals interacts with the surface-modified PEAI molecules via chemical bonds. At the same time, it can be found that the signal from metal Pb (Pb 0) is significantly suppressed, which suggests that there exist fewer lead vacancies inside and on the surface of the dually modified perovskite film [41]. Similarly, the orbital binding energies of I 3d$_{5/2}$ and 3d$_{3/2}$ also move from 619.35 eV and 630.86 eV in the reference sample to the higher-energy region of 619.45 eV and 630.94 eV of the joint passivation sample. By calculation, the corresponding binding energy from Pb-I was slightly reduced by 0.07 eV. This is attributed to Cs$^+$ and Br$^-$ ions in the CsPbBr$_3$ QDs entering the MAPbI$_3$ lattice to form a more stable Cs$_x$MA$_{1-x}$PbI$_{3-y}$Br$_y$ perovskite structure [42,43]. A few amounts of Pb-Br bonds substitute for Pb-I bonds, which thus weaken the binding energy of Pb-I. Overall, the stable perovskite structure, fewer lead ion defects, and surface-modified PEAI will be beneficial for improving the stability of PFs and PSCs.

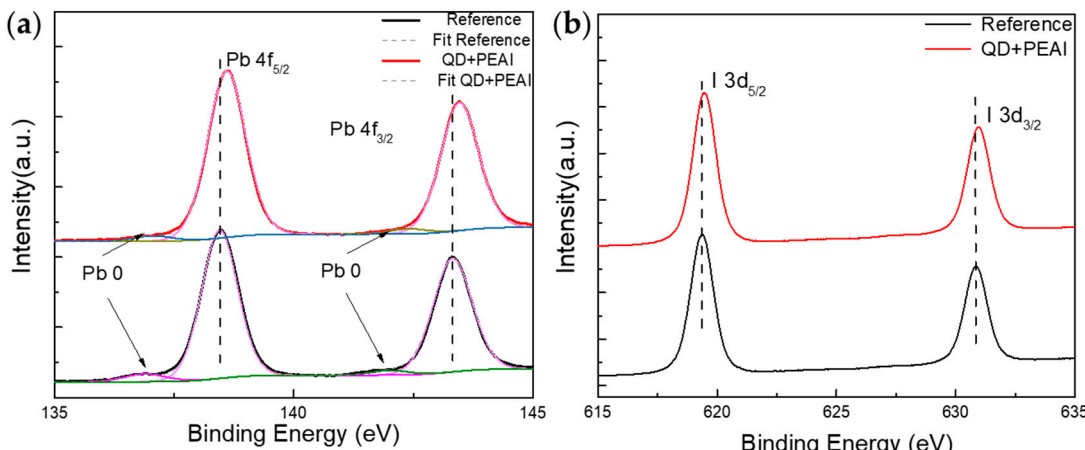

**Figure 6.** XPS spectra of reference sample and perovskite films jointly modified with CsPbBr$_3$ QDs + PEAI: (**a**) Pb 4f and (**b**) I 3d.

### 3.4. Photovoltaic Properties of PSCs Jointly Modified by CsPbBr₃ QDs and PEAI

To directly demonstrate the superior performances of jointly modified PSCs, five kinds of PSCs were fabricated according to the structure shown in Figure 7a, and their photovoltaic characteristics were analyzed. The detailed photovoltaic parameters are listed in Table 1. Figure 7b,c show the J-V scan curves and monochromatic incident photon-to-electron conversion efficiency (IPCE) data of PSCs that were unmodified, modified only by CsPbBr₃ QDs, and jointly modified by CsPbBr₃ QDs and PEAI. PSCs only modified with CsPbBr₃ QDs acquired a PCE of 20.08% with a $V_{oc}$ of 1.13 V, $J_{sc}$ of 23.20 mA/cm², and FF of 76.6% when the optimized CsPbBr₃ QDs concentration of 2 μg/mL was doped, which is markedly higher than the 17.21% of the reference device. The enhanced photovoltaic properties benefit from the obtained stable $Cs_xMA_{1-x}PbI_{3-y}Br_y$ perovskite film with the large grains and low bulk defects which improve light absorption and suppress carrier recombination inside perovskite crystals. PSCs with dual modification of CsPbBr₃ QDs and PEAI can further improve PCE and show the best value of 21.04% with $V_{oc}$ increased to 1.15 V, $J_{sc}$ to 23.30 mA/cm², and FF up to 78.6%. This is because PEAI molecules uniformly cover the grain boundary and interface of the perovskite layer, further fill the ion vacancies, slightly adjust the surface morphology of the film, and effectively block the direct contact between HTL and ETL at the perovskite grain boundary. Correspondingly, the integrated current value of the jointly modified PSC also increases from 19.4 mA/cm² of the reference cell to 22.6 mA/cm², which matches with the current density results obtained from the J-V scanning curve.

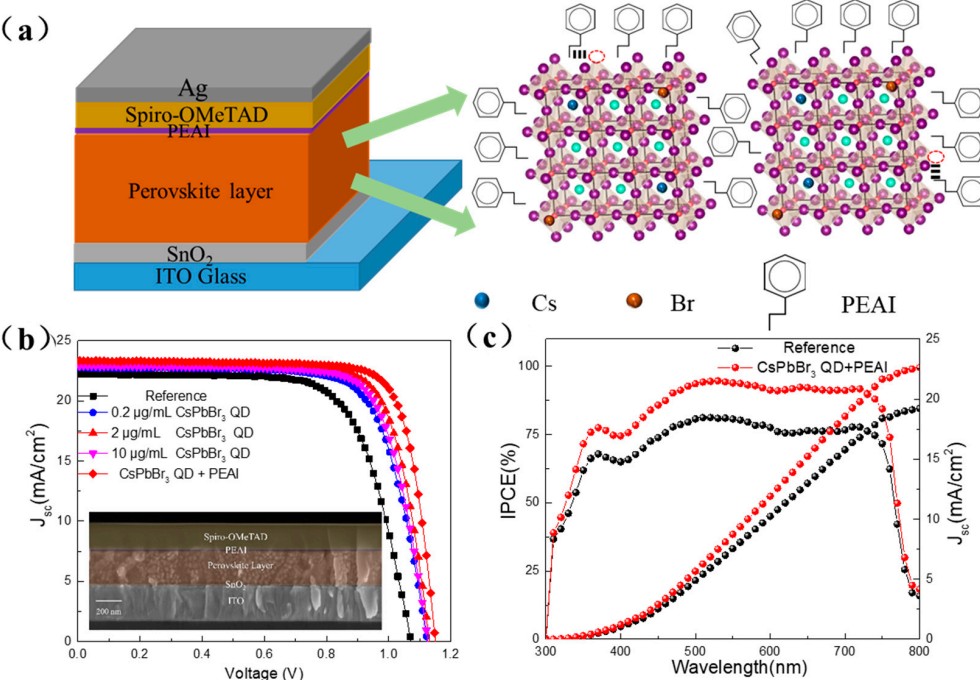

**Figure 7.** (**a**) Structural diagram of PSCs, schematic diagram of perovskite grains jointly modified with CsPbBr₃ QDs and PEAI, (**b**) J-V curves of PSCs modified at different conditions, and (**c**) IPCE patterns.

**Table 1.** The photovoltaic performances of PSCs obtained by different modification methods.

| Sample | Voc (V) | Jsc (mA/cm$^2$) | FF (%) | PCE (%) |
|---|---|---|---|---|
| Reference | Champion 1.07<br>Average 1.06 ± 0.03 | 22.4<br>21.9 ± 0.82 | 71.6<br>70.1 ± 1.7 | 17.21<br>16.3 ± 1.1 |
| 0.2 μg/mL CsPbBr$_3$ QDs | Champion 1.11<br>Average 1.09 ± 0.02 | 22.7<br>22.4 ± 0.52 | 72.3<br>70.9 ± 1.7 | 18.22<br>17.6 ± 0.6 |
| 2 μg/mL CsPbBr$_3$ QDs | Champion 1.13<br>Average 1.11 ± 0.02 | 23.2<br>22.8 ± 0.55 | 76.6<br>75.8 ± 1.2 | 20.08<br>19.7 ± 0.5 |
| 10 μg/mL CsPbBr$_3$ QDs | Champion 1.12<br>Average 1.11 ± 0.02 | 22.8<br>22.5 ± 0.47 | 73.1<br>72.2 ± 1.2 | 18.67<br>18.1 ± 0.6 |
| CsPbBr$_3$ QDs +PEAI | Champion 1.15<br>Average 1.14 ± 0.01 | 23.3<br>22.9 ± 0.56 | 78.6<br>77.2 ± 1.3 | 21.04<br>20.3 ± 0.7 |

The defect density of PFs can be calculated by the space charge limit current (SCLC) curves of electron-only devices modified by different methods. As shown in Figure 8a, the structure of electron-only devices is an ITO/SnO$_2$/perovskite layer with different modified methods/PC61BM/Ag. The corresponding SCLC curves are exhibited in Figure 8b. The voltage cross-points of extension lines from the Tangent of Ohm area and trap-filled area are defined as the trap-filling limit voltage (V$_{TFL}$). The V$_{TFL}$ values of the reference, modified only by CsPbBr$_3$ QDs, and jointly modified by CsPbBr$_3$ QDs and PEAI perovskite devices are 0.564 V, 0.341 V, and 0.221 V, respectively. The reduced V$_{TFL}$ means that defects of the perovskite layer have greatly been suppressed after modification, which can be further proven by the calculated defect density according to the following Equation (3):

$$N_t = \frac{\varepsilon\varepsilon_0 V_{TFL}}{eL^2} \tag{3}$$

where L is the thickness (350 nm), $\varepsilon$ and $\varepsilon_0$ are the dielectric constant and the vacuum permittivity of the perovskite layer, and e is the charge quantity of an electron (1.6 × 10$^{-19}$ C) [43,44]. The defect density (N$_t$) of the doubly modified perovskite layer is only 0.71 × 10$^{16}$ cm$^{-3}$, which is obviously smaller than 0.97 × 10$^{16}$ cm$^{-3}$ of the solely QD-modified sample and 1.65 × 10$^{16}$ cm$^{-3}$ of the reference perovskite layer, which supports the best performance of PSCs in Figure 7.

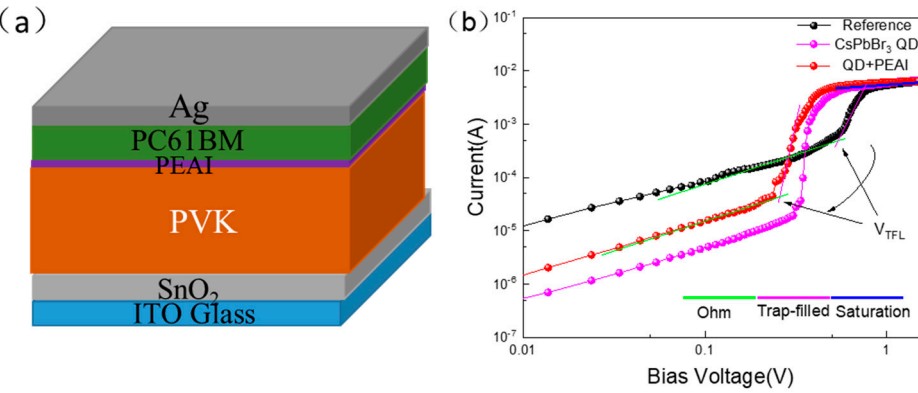

**Figure 8.** (**a**) The structure of the electron-only device and (**b**) SCLC of perovskite film modified by different methods.

The dark J-V curves and electrochemical impedance spectroscopy (EIS) of three kinds of PSCs were also measured and are given in Figure 9. It is evident that the dark current density of the CsPbBr$_3$ QD-modified PSCs has decreased by nearly two orders of magnitude compared to the reference device, while the PSCs jointly modified with QDs and PEAI have the lowest dark current density. The corresponding recombination resistance is 3986 Ω, which is larger than the values of 2987 and 1156 Ω for the QD-modified and reference PSCs, respectively. These results further support that the joint modification of QDs and PEAI can reduce carrier recombination at the interface and improve PCEs.

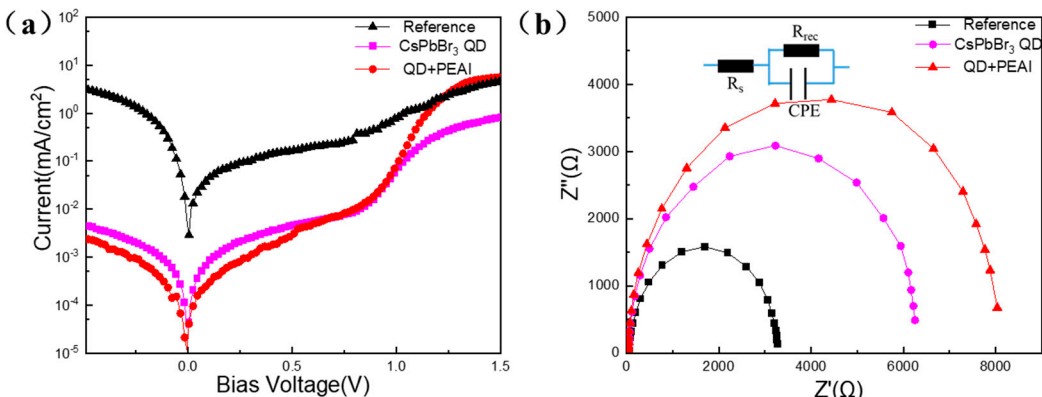

**Figure 9.** (**a**) Dark J-V curves and (**b**) EIS (dark condition, bias voltage -0.85 V) of PSCs modified at different conditions.

Figure S3 shows TPV and TPC decay curves of the original and jointly modified PSCs. The lifetime of carriers is defined as the time of the cell voltage drop to the 1/e of the initial value [45]. It can be seen in Figure S3a that the lifetime of the reference cell is 0.087 ms, while after dual modification, it is dramatically enhanced to 0.512 ms. The slower voltage decay suggests that the carrier recombination has been limited because of the dual passivation and blocking effect of QDs combined with PEAI. Simultaneously, the faster photocurrent decay for jointly modified PSCs (23.1 μs relative to 35.5 μs in the original cell) in Figure S3b indicate that carriers can be more easily extracted and transported from the perovskite layer to the carrier transport layer. Furthermore, the energy alignment of the reference, CsPbBr$_3$ QD-, and PEAI-modified PSCs are given in Figure S4. The doping of QDs has a negligible impact on the energy levels of the MAPbI$_3$ reference layer [22]. However, after PEAI modification, a thin insulating layer of PEAI molecules is formed at the interface of the perovskite film, creating a high-energy-level barrier. According to the hole tunneling equation (Supporting Information) [30], under the influence of the built-in electric field, it is more likely for the holes to tunnel from the valence band of the perovskite layer to the valence band of the Spiro-OMeTAD hole transport layer, thereby promoting the transport and collection of hole carriers. Simultaneously, the insulation effect reduces the probability of electron–hole recombination at the interface between the conduction band of the perovskite and the valence band of the Spiro-OMeTAD layer [30,46], ultimately enhancing the optoelectronic performance of the co-modified PSCs.

### 3.5. Stabilities of PSCs Jointly Modified by CsPbBr$_3$ QDs and PEAI

Figure 10a,b present the hydrophilic contact angle test for the unmodified and jointly modified PFs. The contact angles are 34.5° and 48.2°, respectively, indicating that the spin-coated PEAI molecular layer enhances the hydrophobicity of PFs, which can better suppress the erosion of water vapor on perovskite films and improve the ambient stability of the final devices. Furthermore, the steady-state maximum power output curves of two kinds of unpacked PSCs were recorded at different bias voltages for 120 s under the ambient conditions of 20 °C and relative humidity of 40% in Figure 10c. The QD and PEAI jointly modified PSCs show not only a higher power output of 20.56% but also a better stability of current output. After aging for 30 days, the normalized PECs of the jointly modified cell still maintain 75% of the original PCE, while the reference PSCs only retain 59% of the initial value (Figure 10d). Furthermore, the V$_{OC}$, FF, and J$_{SC}$ of the jointly modified PSCs also show the relatively gentle changes compared to the reference cell (Figure S5). The better crystallinity and hydrophobicity properties, the more compact and smoother surface, and the reduced defect density of the perovskite layer can contribute to better light and ambient stability for the QD and PEAI jointly modified PSCs [47].

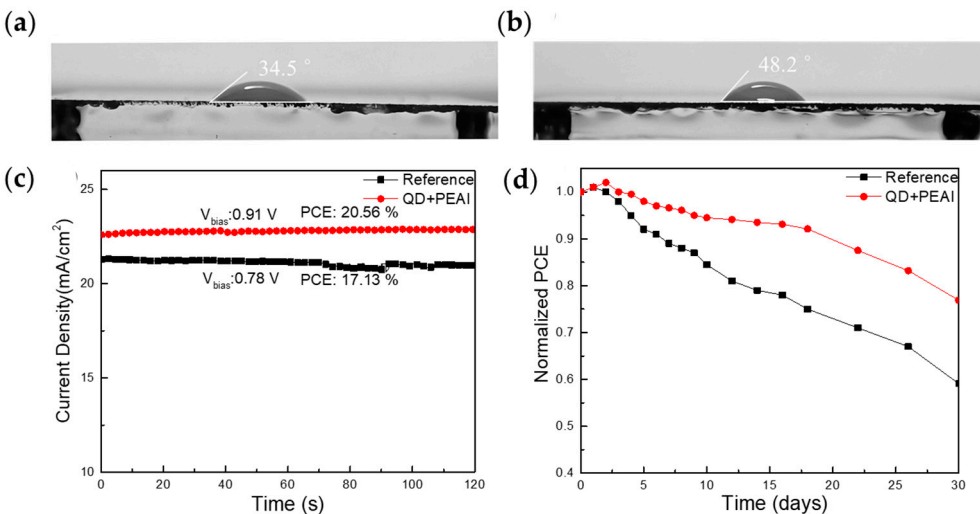

**Figure 10.** Contact angles of (**a**) the reference and (**b**) CsPbBr$_3$ QD and PEAI jointly modified PFs (deionized water); (**c**) steady-state maximum current output under 100 mW/cm$^2$ simulating sunlight at the bias voltage of 0.78 V for the original PSCs and 0.91 V for the jointly modified cells; (**d**) aging curves of unpacked PSCs in air condition with a relative humidity of 40% and at RT.

## 4. Conclusions

In summary, we have demonstrated that the dual-protection strategy via a bulk dopant with CsPbBr$_3$ QDs and surface modification with PEAI has enabled us to enhance the efficiency and stability of PSCs. The doped CsPbBr$_3$ QDs acted as condensation nuclei to promote the nucleation and grain growth of MAPbI$_3$ films. Simultaneously, Br$^-$ and Cs$^+$ entered into the inner crystal lattice of MAPbI$_3$ to replace MA$^+$ and I$^-$ and form dense and stable Cs$_x$MA$_{1-x}$PbI$_{3-y}$Br$_y$ PFs with low defects and large grains. Furthermore, the modified PEAI molecules not only formed a uniform ultrathin layer to block the direct contact of ETL and HTL at the perovskite grain boundary, but also filled the ion vacancies and passivated defects on the surface, thereby suppressing the recombination of carriers. High-quality PFs, excellent interface contacts, and hydrophobic properties together promoted the transport and extraction of carriers at the interface between the perovskite layer and HTL. As a result, the fabricated PSCs with dual modification exhibited an improved PCE of 21.04% with a V$_{oc}$ of 1.15 V, J$_{sc}$ of 23.30 mA/cm$^2$, and FF of 78.6%. The corresponding better light and ambient stability has been achieved.

**Supplementary Materials:** The following supporting information can be downloaded at: https://www.mdpi.com/article/10.3390/coatings14040409/s1, Figure S1: AFM images and grain size statistics of perovskite films modified with different conditions: (a) reference, (b) doped with 2 µg/mL CsPbBr$_3$ QDs, (c) jointly modified with CsPbBr$_3$ QDs and PEAI.; Figure S2: EDS mappings of PFs jointly modified with CsPbBr$_3$ QDs and PEAI: (a) SEM pattern, (b) C element, (c) Pb element, (d) Br element, (e) I element, (f) the combination of SEM and mappings. Table S1: The parameters of (110) diffraction peak of PFs modified at different conditions; Table S2: Simple comparison of PCE MAPbI$_3$ perovskite solar cells. Figure S3: (a) Dark J-V curves and (b) EIS (dark condition, bias voltage −0.85 V) of PSCs modified at different conditions. Figure S4: TPV (a) and TPC (b) decay curves of original and jointly modified PSCs. Figure S5: Aging curves of normalized V$_{OC}$ (a), FF(b) and J$_{SC}$ (c) in air condition with a RH of 40% and RT (unpacked). References [22,48–54] are cited in the supplementary materials.

**Author Contributions:** Conceptualization: S.Z. and T.X.; Methodology: S.Z. and N.H.; Investigation: S.Z. and S.H.; Data curation: S.Z. and H.L.; Formal analysis: S.Z.; R.C. and M.Q.; Resources: M.Q.; B.L. and X.C.; Writing—Original Draft: S.Z.; Writing—Review and Editing: X.C., T.X. and H.L.; Supervision: H.L. and X.C. All authors have read and agreed to the published version of the manuscript.

**Funding:** This work was supported by the National Natural Science Foundation of China (No. 12274136), Shanghai Municipal Natural Science Foundation (No. 19ZR1415400, 18ZR1411000 and 18ZR1411900), Chongqing Municipal Natural Science Foundation (No. 33606015, Chongqing Key Laboratory of Precision Optics, Chongqing Institute of East China Normal University, Chongqing 401120, China), and Joint Institute of Advanced Science and Technology, East China Normal University, Shanghai 200062, People's Republic of China (No. 40500-20105-222053).

**Institutional Review Board Statement:** Not applicable.

**Informed Consent Statement:** Not applicable.

**Data Availability Statement:** Data are contained within the article and Supplementary Materials.

**Conflicts of Interest:** The authors declare no conflicts of interest.

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
