# Peer review of "Dual Strategy Based on Quantum Dot Doping and Phenylethylamine Iodide Surface Modification for High-Performance and Stable Perovskite Solar Cells"

_coatings, doi:10.3390/coatings14040409_

Round 1
Reviewer 1 Report
Comments and Suggestions for Authors
Authors have demonstrated improvement the stability and efficiency of PSCs by combining CsPbBr3 QDs and PEAI. Although, either the CsPbBr3 QDs or PEAI is not new method, their combinations are interesting. PSCs applied both CsPbBr3 QDs and PEAI exhibited the relative high efficiency (21.04%) and good stability (retain 75% of their initial efficiency after 30 days at humidity of 40%).
I think the manuscript could be published on Coatings after minor revision, as following:
1. in figure 3e, authors should describe the concentration QDs. Furthermore, the average size of all perovskite crystal should be provided and discussed.
2. in figure 5d, authors should add the TRPL spectra of pervoskite/QD (and calculate the life time in this case) to identify the role of QD layer.
3. in figure 9d, author should add the dependence of Jsc, Voc and FF on aging time and discuss on it.
Reviewer 2 Report
Comments and Suggestions for Authors
In the paper authors report a dual-modification strategy via incorporating CsPbBr3 QDs into MAPbI3 perovskite bulk and capping the interface of the perovskite/hole transport layer (HTL) with phenylethylamine iodide (PEAI) to improve perovskite crystallinity and interface contact properties to acquire high-quality PFs with fewer defects.
The authors demonstrated the benefits from growth control and effective suppression of both bulk and interface carrier recombination, the resulting devices show a greatly improved photoelectric conversion efficiency (PCE) from 25 17.21% of the reference cells to 21.04 % with a champion Voc of 1.15 V, Jsc of 23.30 mA/cm2 and fill 26 factors (FF) of 78.6 %.
However, in the present form, the paper is not suitable for publication in Coatings journal. It certainly needs major revision for the following reasons:
In Fig. 2 shows the absorption spectrum of CsPbBr3 QDs; However, it is not correct to show the absorption spectrum as a function of intensity versus wavelength. The authors need to add another Y scale and label it. What was the thickness of the samples when measuring the absorption spectrum?
After point 3.2. shown in Fig. 3 the authors need to change the order by adding a description above the picture. The scale bar is hard to see. What is the grain size based on SEM results? The authors need to show the particle size distribution based on the SEM results.
It is necessary to submit standard cards for all XRD results, confirming that they belong to the materials being studied (Fig. 2, 4).
In Fig. 5 it is necessary to add approximation curves for luminescence decay kinetics. Based on Fig. 5d it is difficult to say that the decay kinetics can be fitted as a bi-exponential function. This point of research needs to be clarified.
What was used as a reference sample that compare with CsPbBr3 QDs device?
From Fig. 7, the meaning of the arrows pointing to the same perovskite structures is unclear, but they come from SnO2 and Spiro-OMeTAD. It is necessary to clarify this point.
There is no charge transport scheme with energy levels diagram for prepared PCSs. It is necessary to supplement the work with these results and relevant discussion.
To determine the mechanisms of transport and recombination of the charge carriers, the impedance spectra results of fabricated cells should be conducted and included in the manuscript.
Reviewer 3 Report
Comments and Suggestions for Authors
The authors presented article titled ‘Dual Strategy Based on Quantum Dots Doping and PEAI Surface Modification for High-Performance and Stable Perovskite Solar Cells’ has tried to improve the performance but it lacks novelty and just incremental the present version and recommend reconsidering after major revision.
A few suggestions the authors can incorporate to improve the manuscript.
1. The utilization of phenylethylamine iodide is noted as providing only incremental improvements in performance, with no assessment of durability for stable output.
2. The manuscript lacks clarification regarding the two decay channels observed in the lifetime data. Moreover, the rationale behind averaging lifetime values and the cause of the observed 128.3 ns lifetime remain unexplained.
3. The reason for the observed increase in photoluminescence (PL) intensity following PEI coating is not adequately addressed.
4. The manuscript fails to include a comparison between the reported literature and the claimed increase in efficacy. I suggest creating a table for easy comparison of efficiencies.
Comments on the Quality of English LanguageEnglis is ok and readable
Round 2
Reviewer 3 Report
Comments and Suggestions for Authors
The authors have included all the comments. I have no reservations about the recommendation for the present version.